# Estimation of the Total Soil Nitrogen Based on a Differential Evolution Algorithm from ZY1-02D Hyperspectral Satellite Imagery

**Rongrong Zhang [1], Jian Cui [2,3], Wenge Zhou [1], Dujuan Zhang [4], Wenhao Dai [1], Hengliang Guo [4] and Shan Zhao [1,\*]**

[1] School of Geoscience and Technology, Zhengzhou University, Zhengzhou 450001, China; 202012562017389@gs.zzu.edu.cn (R.Z.); zhouge2022@gs.zzu.edu.cn (W.Z.); 202022562017409@gs.zzu.edu.cn (W.D.)

[2] Henan Institute of Geological Survey, Zhengzhou 450001, China; 2021811@zzuli.edu.cn

[3] National Engineering Laboratory Geological Remote Sensing Center for Remote Sensing Satellite Application, Zhengzhou 450001, China

[4] National Supercomputing Center in Zhengzhou, Zhengzhou University, Zhengzhou 450001, China; duduzdj@zzu.edu.cn (D.Z.); guohen-gliang@zzu.edu.cn (H.G.)

\* Correspondence: zhaoshan4geo@zzu.edu.cn; Tel.: +86-1370-0875-423

**Abstract:** Precise fertilizer application in agriculture requires accurate and dependable measurements of the soil total nitrogen (TN) concentration. Henan Province is one of the most important grain-producing areas in China. In order to promote the development of precision agriculture in Henan Province, this study took the high-standard basic farmland construction area in central Henan Province as the research area. Using single-phase images acquired from the ZY1-02D satellite hyperspectral sensor on 28 January 2021 (with a spatial resolution of 30 m × 30 m, a spectral range that covered 400–2500 nm, and a revisit period of 3 days) for spectral reflectance transformation and feature spectral band extraction. Based on multiple representation models, such as multiple linear regression, partial least squares regression, and support vector machine (SVM), all bands, feature bands, feature band combinations, and differential evolution (DE) algorithms were used to extract the secondary characteristic variables of the combination of characteristic bands, which were used as model inputs to estimate the content of TN in the study area. It was found that (1) the spectral reflectance transformation could help to improve the accuracy of prediction by reducing the interference from noise in the model, but the optimal spectral transformation method differed between different models and even between the training and test sets of the same model; (2) the estimation accuracy of the TN content model based on the minimum shrinkage and feature selection operator of the feature band was usually better than that of the full band, the feature combination band contained more effective information related to the TN content, and the combination of the DE algorithm and the SVM model achieved a better estimation accuracy for secondary feature extraction and TN content estimation of the feature combination band; and (3) ZY1-02D hyperspectral satellite data have the potential for the dynamic and non-destructive monitoring of regional TN content.

**Keywords:** soil total nitrogen; ZY1-02D/AHSI hyperspectral; feature selection; model estimation

## 1. Introduction

Soil total nitrogen (TN) is a fundamental indicator of soil fertility and a critical factor in plant growth and development [1,2]. The soil's abundance of or deficiency in nitrogen will directly impact crop growth and yield [3]. Therefore, the dynamic, large-scale, and accurate estimation of TN content is a significant measure that can be used to guide agricultural field fertilization schemes and crop growth status monitoring [4].

The traditional method for determining TN is to gather soil samples from different sites for laboratory chemical analysis, after which the TN distribution is obtained using

spatial interpolation. In order to guarantee the precision of the interpolation results, this method frequently calls for collecting numerous soil samples, which is time-consuming and invasive [5,6]. Recent research has demonstrated that satellite-based hyperspectral images offer excellent spectral and spatial resolution, enabling dynamic, effective, and precise predictions of soil components [7–9]. The ZY1-02D satellite carries the Advanced Hyper Spectral Imager (AHSI), which can acquire 166 bands of different wavelengths between 400 and 2500 nm (covering visible to shortwave infrared), with an image strip width of 60 km [10]. Recent research has demonstrated that the ZY1-02D/AHSI satellite, China's first hyperspectral satellite for civil usage, has promising application possibilities, and images obtained using the AHSI sensor can be used to estimate the component composition of the soil [11,12].

Hyperspectral data are rich in spectral information that may be used to identify changes in soil characteristics and offer precise estimations of the elemental content in the soil [9,12,13]. However, there are several problems with using hyperspectral data to estimate soil nutrient contents, such as the fact that unprocessed original hyperspectral data typically contain a lot of redundant spectral information because they have a lot of spectral channels with high spectral resolution [14]. The recent research on hyperspectral-based soil content estimation has mainly included the processing of spectral data through reflectance transformation, spectrum feature selection, and the formulation of estimation models. Reflectance transformation processing can enhance feature bands and facilitate the rejection of noise [15]. Therefore, choosing an appropriate reflectance transformation technique is crucial in guaranteeing the model's precision [16].

Studies have shown that spectral data transformation methods, such as the inverse reflectance [17], natural logarithm of the reflectance [18], and first-order derivative reflectance [19], can enhance the characteristic bands and improve model accuracy [20,21]. In addition, some scholars have studied spectral information and found that the selection of an appropriate spectral feature band can reduce data redundancy, simplify the model, improve model accuracy, and lead to better estimation results [22,23]. Spectral feature extraction has been carried out using methods such as the differential evolutionary (DE) technique and the least absolute shrinkage and selection operator (LASSO) [4,24]. However, most studies on the extraction of characteristic bands of soil composition information are based on original spectral data or single spectral transformation data without considering the application potential of characteristic spectral combination data in soil composition information model estimation. Currently, machine learning models, such as support vector machines (SVMs), back propagation neural networks, and random forests [25–28], as well as linear models, such as multiple linear regression (MLR) and partial least squares regression (PLSR) [29–31], are commonly used to estimate the soil component content. Scholars demonstrated that SVM models produce more precise predictions than the RF and PLSR models [32], and the SVM model has good stability and versatility in solving nonlinear problems.

However, the current hyperspectral data used for TN content estimation are mainly based on hyperspectral data obtained by laboratories and ground platforms [33], and there is still a lack of research on TN content estimation methods based on satellite platform hyperspectral data, especially the new ZY1-02D hyperspectral data. Summarizing the previous research results, this study established a set of TN content estimation processes and methods based on ZY1-02D hyperspectral remote sensing data from spectral reflectance transformation processing, spectral band selection, and model construction methods. In order to make full use of the computational spectral data obtained by spectral reflectance transformation and feature selection, this study also proposed the use of characteristic spectral combination data for the estimation of TN content. Additionally, this study aimed to tackle the problem of possible redundant information in the characteristic spectral combination data, and proposed performing secondary feature selection on the characteristic spectral combination data to remove invalid variables and improve the accuracy of model estimation.

In this work, by comparing previous studies, the ZY1-02D hyperspectral data were transformed using four widely used techniques: the original reflectance, the inverse reflectance, the natural logarithm of the reflectance, and the first-order derivative reflectance. The LASSO technique was used to extract the primary spectral characteristics for individual spectral reflectance transform data, and a first attempt was made at secondary feature extraction using the DE algorithm for LASSO feature spectral band combination data. Finally, based on the chosen characteristics, the MLR, PLSR, and SVM models were used to estimate the TN content based on the selected characteristics, and the best model was selected to map the TN content. These were the key goals: (1) to investigate the best spectral reflectance transformation methods for different models; (2) to compare the estimation of all bands of a single spectral reflectance transformation with the LASSO feature bands; and (3) to investigate the model estimate capabilities of secondary feature selection for a combination of LASSO feature bands using the DE technique.

## 2. Materials and Methods

### 2.1. Study Area

In this study, the construction area of high-standard basic farmland in central Henan Province, China, which is located between 34°16′ and 34°40′ N and 113°20′ and 113°54′ E (Figure 1), was used as the study area. The terrain is relatively elevated in the east, with hills in the center and mountains in the west, and with elevations ranging from 78 to 787 m. The area is one of the major grain-producing areas in Henan Province, with an average annual temperature of 14.4 °C and a mild monsoon environment. The primary crops grown there are soybeans, maize, and wheat.

### 2.2. Soil Sample Acquisition and TN Content Determination

Using the five-point sampling technique (collect soil samples at the center and four corners of a rectangular area of 30 m × 30 m and mix the five samples evenly as the final collected soil sample), 595 soil samples were gathered at depths of 0–20 cm in the research area in March 2021 and their positions were recorded (Figure 1c). The soil samples were sieved to remove crop roots, gravel, and other contaminants and allowed to dry in a chamber, and then the total nitrogen content of the soil samples was determined using an automatic Kjeldahl nitrogen analyzer [34]. At the same time, the amounts of heavy metals (cadmium, mercury, arsenic, lead, chromium, copper, nickel, and zinc) and other soil nutrients (organic matter, total phosphorus, total potassium, and pH value) in the soil samples were also determined.

### 2.3. ZY1-02D/AHSI Remote Sensing Image Collection and Pre-Processing

Soil sampling was conducted in March 2021, which is when winter wheat was the dominant crop in the study area. This is the time when young winter wheat seedlings turn green and begin to grow. Considering the influence of the soil sampling time and clouds on the image quality, hyperspectral remote sensing images acquired by the ZY1-02D/AHSI sensor on 28 January 2021 (Figure 1c) were the closest high-quality images to the soil sampling time, and they were taken at a time when wheat and other crops were growing slowly in winter and the plants were short and did not cover the ground. The soil surface reflectance obtained with hyperspectral satellites during this period was less affected by crops and was similar to the spectral reflectance of bare soil; therefore, this study was based on the images from this period for the estimation of TN content in the study area. The ZY1-02D/AHSI sensor's parameters are listed in Table 1. The satellite operated in a Sun-synchronous orbit and acquired hyperspectral images in 166 spectral channels over a swath width of 60 km. The spectrum range covered was between 400 and 2500 nm. The resulting images comprised 90 channels between 1055 and 2500 nm with a resolution of 20 nm and 76 spectral channels between 400 and 1040 nm with a spectral resolution of 10 nm.

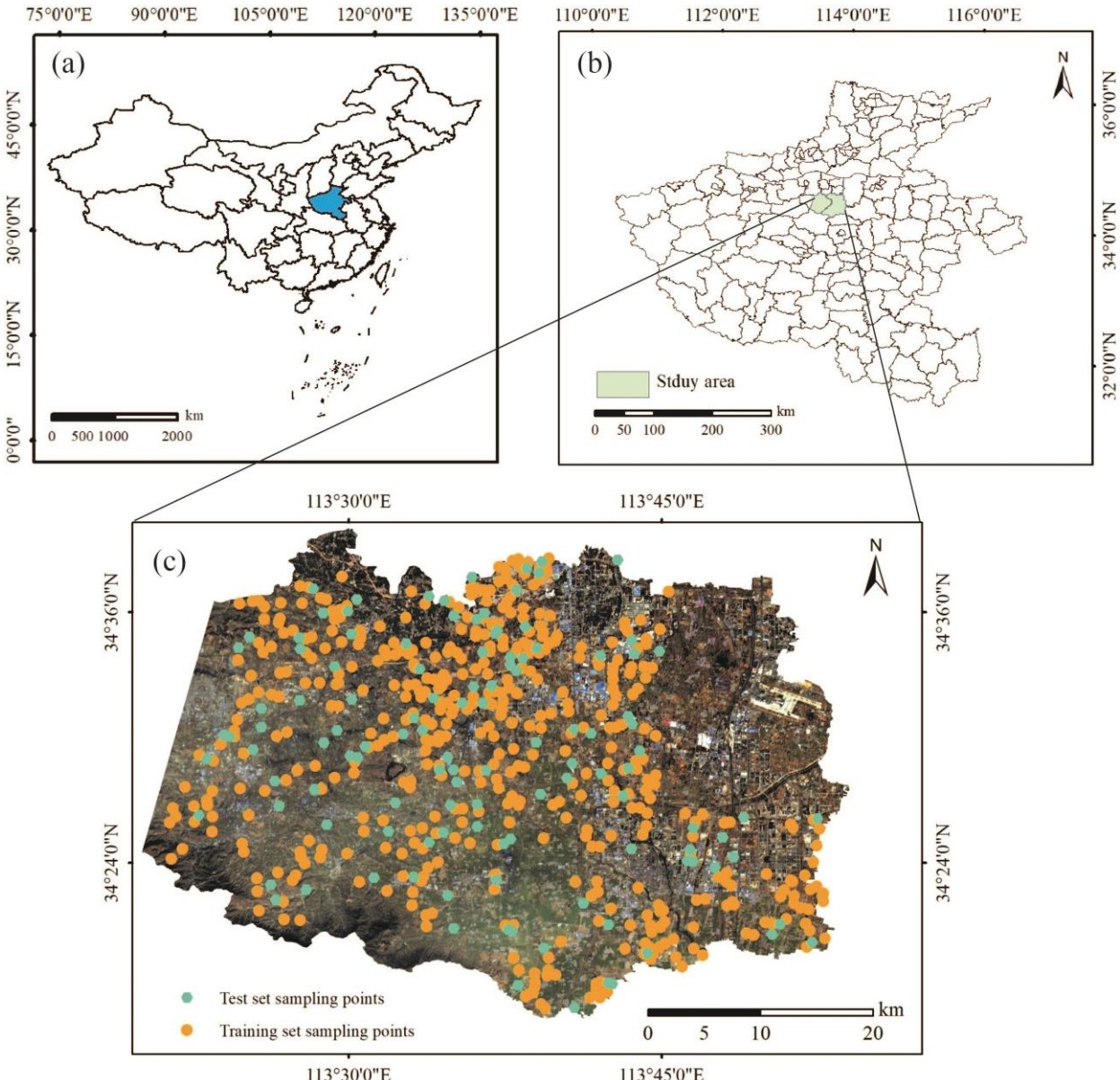

**Figure 1.** Maps of the research site. (**a**) Map of China, where the blue area is Henan Province; (**b**) map of Henan Province; (**c**) hyperspectral ZY1-02D/AHSI image of the research area, together with the locations of the soil sampling sites (in this study, only the reflectance data of satellite images in cultivated areas were extracted for experimentation, excluding buildings and other areas).

**Table 1.** Device parameters of ZY1-02D/AHSI.

| Items | Parameters |
| --- | --- |
| Date of launch | 12 September 2019 |
| Spectral bands | 76 (VNIR), 90 (SWIR) |
| Spectral range (nm) | 400–2500 |
| Spectral resolution (nm) | 10 (VNIR), 20 (SWIR) |
| Spatial resolution (m) | 30 |
| Swath width (km) | 60 |
| Revisit cycle (d) | 3 |

The images were pre-processed to improve their quality and reduce the effects caused by weather and atmospheric factors. The images were radiometrically and atmospherically corrected and orthorectified using ENVI v5.3. Subsequently, to determine the reflectance

value of the hyperspectral pictures corresponding to the coordinates of the soil sampling location, ArcGIS v10.5 was utilized. In this study, three spectral bands that overlap with VNIR and SWIR and four bands near 1400 nm and 1900 nm that are strongly influenced by atmospheric water absorption (two bands near 1400 nm: B100 and B101 at 1391.65 nm and 1408.49 nm, and two bands near 1900 nm: B130 and B131 at 1896.11 nm and 1912.97 nm) were used, for a total of 159 experimental spectral bands.

### 2.4. Methodology

In this study, the TN estimation using satellite hyperspectral imagery consisted of the following three main steps: (1) data acquisition and pre-processing, as well as preparation of soil sampling points and remote sensing image data (as described in Sections 2.2 and 2.3 above); (2) data processing, spectral reflectance transformation, and feature extraction; (3) and TN content model estimation and mapping. The experimental approach for this investigation is shown in a flow chart in Figure 2.

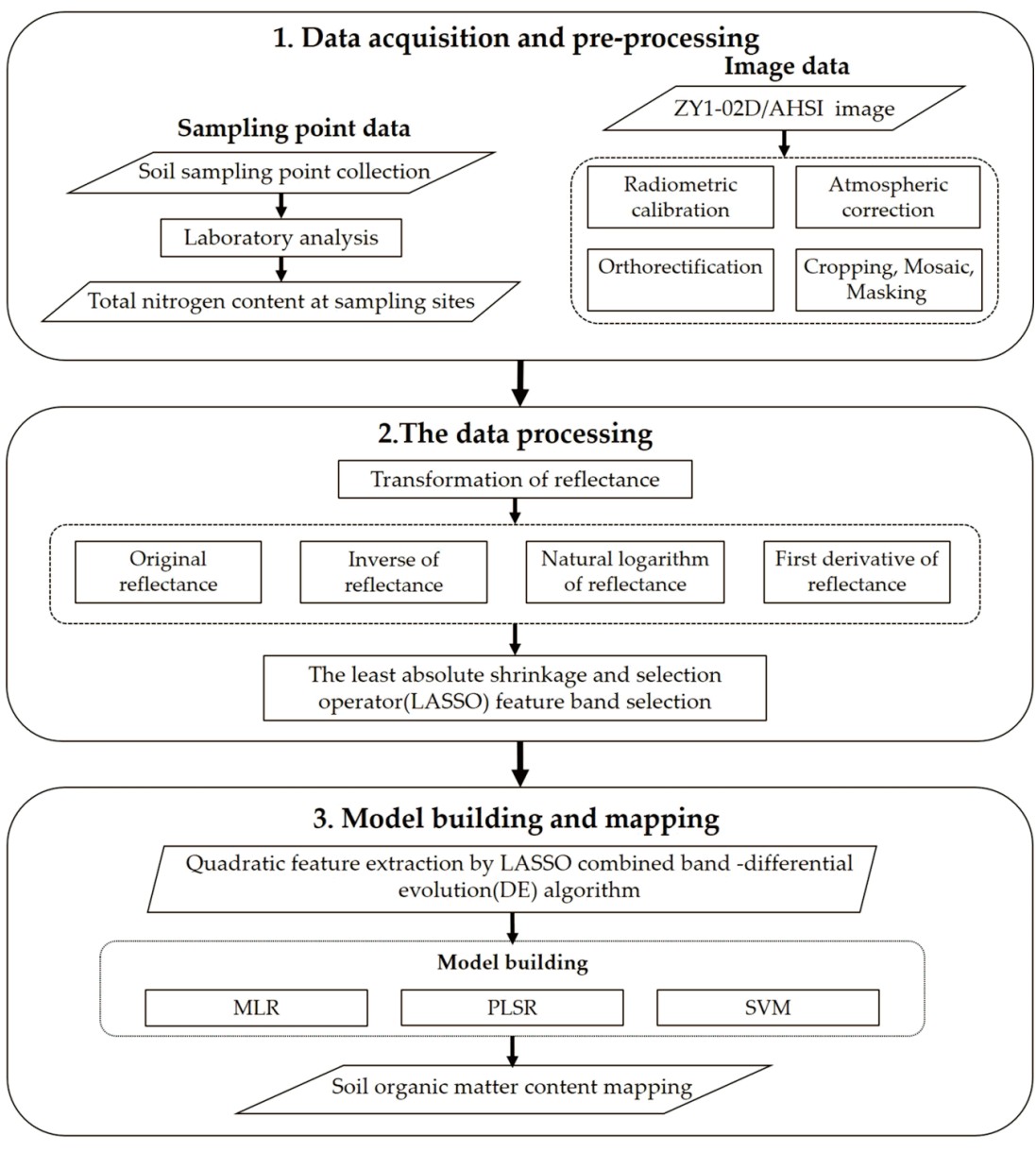

**Figure 2.** Flow chart of the experimental steps.

2.4.1. Spectral Reflectance Transformation

To identify the sensitive links between TN content and spectral reflectance and emphasize distinctive spectral bands, it is crucial to transform the spectral data. In this study, the inverse reflectance (IR), the natural logarithm of the reflectance (NLR), and the first-order derivative reflectance (FDR) were used to transform the original reflectance (OR) values. Spectral transformation processing can increase spectral sensitivity and aid in increasing models' predictive ability [35]. The equations for the different transformations are shown below:

$$IR(\lambda_i) = \frac{1}{\lambda_i} \tag{1}$$

$$NLR(\lambda_i) = LnR(\lambda_i) \tag{2}$$

$$FDR(\lambda_i) = \frac{R(\lambda_{i+1}) - R(\lambda_{i-1})}{2\Delta\lambda} \tag{3}$$

where $\lambda_{i-1}$, $\lambda_i$, and $\lambda_{i+1}$ are the wavelengths of band i and its adjacent bands; $\Delta\lambda$ indicates the interval between two adjacent wavelengths; $IR(\lambda_i)$ is the inverse reflectance of wavelength $\lambda_i$; $NLR(\lambda_i)$ is the natural logarithmic reflectance of wavelength $\lambda_i$; and $FDR(\lambda_i)$ is the first-order derivative reflectance of wavelength $\lambda_i$.

2.4.2. Selection of Spectral Feature Variables

According to previous research, the identification of distinctive spectral characteristics is one of the key techniques for estimating the concentration of soil components using hyperspectral data [36,37]. It is essential to choose the proper feature variables in order to ensure the precision of the model estimation. TN content and spectral reflectance values have a complicated, non-linear relationship. Thus, in this work, the TN content feature bands were selected using non-linear algorithms (LASSO and DE). The LASSO algorithm was used for the initial feature selection of the above four sets of spectral reflectance data. Then, the DE algorithm combined with a predictive model was used to perform secondary feature selection from the feature band combinations selected using LASSO. The LASSO and DE algorithms were implemented using Python 3.8 and are described below.

LASSO is a paradigm-based algorithm in which a 1-paradigm regularization penalty term was introduced on top of the ordinary least squares function to constrain the residual sum of squares [38], which could successfully reduce the dimensionality of the data and resolve sparsity issues with high-dimensional data. To carry out feature selection, the estimation was compressed using a penalty function that compressed the regression coefficients. The coefficients of the less-sensitive variables were adjusted to zero when the total absolute value of the regression coefficients was less than a specific value. The following is a mathematical definition of the LASSO algorithm.

$$\operatorname*{argmin}_{\varrho}\left\{\sum_{i=1}^{n}\left(p_i - \sum_{j=1}^{m} h_{ij}X_j\right)^2\right\}$$
$$\text{subject to} \sum_{j=1}^{m}\left|X_j\right| \le \varepsilon \tag{4}$$

where the numbers n and m denote the samples and variables, respectively; the independent and dependent variables for each sample are denoted by $h_{ij}$ and $p_i$; and $\varepsilon$ and $X_j$ are the critical and specific gravity values.

The DE method, which is a global optimization approach, was initially put forward by R. Storn and K. Price [39]. It is extensively utilized in many domains because of its straightforward structure, implementation, quick convergence, and robustness. Some academics have also employed the method for feature variable selection [40]. The DE algorithm is a population-based heuristic search algorithm. Its fundamental premise is

to produce candidates for selection through individual variations within a population, followed by crossover and selection operations to accomplish population evolution. In order to advance to the next generation and determine the optimal model variable selection, the DE algorithm goes through an evolutionary process, which is depicted in Figure 3 and comprises mutation, crossover, and selection processes.

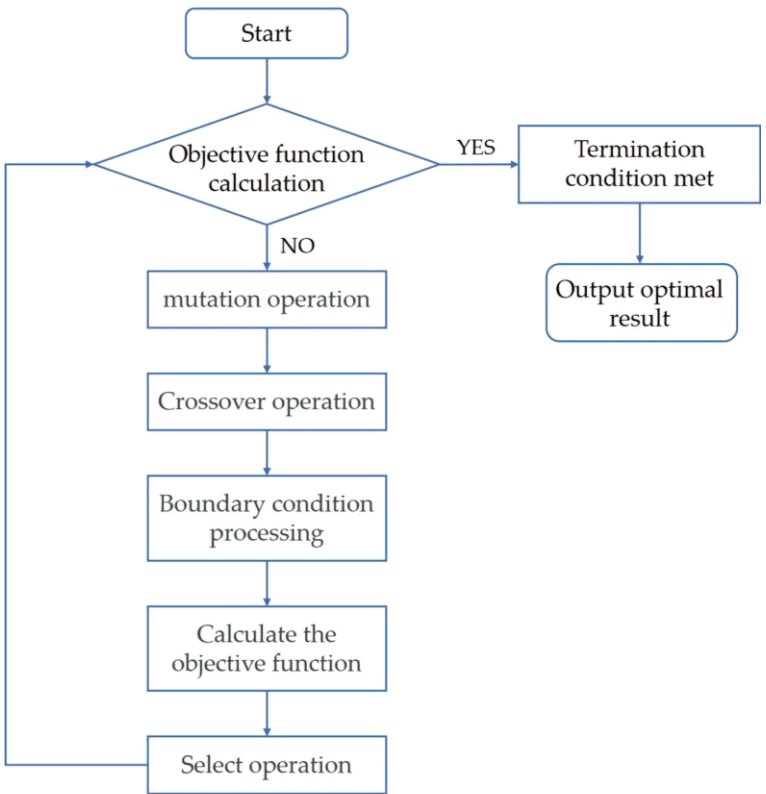

**Figure 3.** DE algorithm's flow chart.

Some of the expressions for the calculation of the DE algorithm are shown below:

(a) Initializing the population: M individuals, each made up of an n-dimensional vector, are created uniformly and randomly in the solution space. The vector and the j-th dimensional value assigned to the i-th individual, respectively, are given in Equations (5) and (6):

$$X_i(0) = (x_{i,1}(0), x_{i,2}(0), x_{i,3}(0), \ldots, x_{i,n}(0))$$
$$i = 1,\ 2,\ 3, \ldots,\ M \tag{5}$$

$$X_{i,j}(0) = L_{j\_min} + \text{rand}(0,1)\left(L_{j\_max} - L_{j\_min}\right)$$
$$i = 1,\ 2,\ 3, \ldots,\ M; j = 1,\ 2,\ 3 \ldots,\ n \tag{6}$$

where $X_i(0)$ is the i-th individual; j denotes the j-th dimension; M denotes the population size parameter; n denotes the optimization dimension; $L_{j\_min}$ and $L_{j\_max}$ are the lower and upper bounds of the j-th dimension, respectively; and $\text{rand}(0,1)$ denotes a random number on the interval [0, 1].

(b) Mutation operation: The DE algorithm implements an individual mutation operation through a difference strategy. Equation (7) shows the vector mutation operation for each individual:

$$W_i(G+1) = X_{c1}(G) + Z(X_{c2}(G) - X_{c3}(G)) \tag{7}$$

where c1, c2, and c3 are random numbers; Z is a scaling factor; and G is the index of individual mutation generations.

(c) Crossover operation: the mutated individuals are subject to the crossover operation shown in formula eight:

$$U_{i,j}(G+1) = \begin{cases} W_{i,j}(G+1) & \text{if rand } (0,1) \leq CR \\ x_{i,j}(G) & \text{otherwise} \end{cases} \tag{8}$$

where CR is the crossover probability.

(d) The operation for selecting the next generation of individuals is shown in Equation (9):

$$X_i(G+1) = \begin{cases} U_i(G+1) & \text{if } (U_i(G+1) \leq f(X_i(G)) \\ X_i & \text{otherwise} \end{cases} \tag{9}$$

where f is the objective function.

### 2.4.3. Construction and Evaluation of the TN Content Estimation Model

The TN content was used as the dependent variable in this research, and the independent variables were the spectral band reflectance values of various forms. Three models, namely, MLR, PLSR, and SVM, were used to describe the relationship between the spectral data and TN content. The models' specific characteristics are listed below.

MLR is a type of regression analysis where the dependent variable is predicted or estimated using a combination of several independent factors. MLR is also widely used as a classical prediction model for predicting soil information content [41,42]. The MLR model has the benefit of being straightforward to construct and simple to apply when compared with other models. However, the MLR model cannot produce a precise forecast of the target variables when there is a non-linear connection between the independent factors and the target variables.

PLSR is a popular multivariate statistical technique for forecasting soil element content using hyperspectral data [43,44], which can address the issue of spectral band covariance. The PLSR approach compares several dependent variables to numerous independent variables in a multivariate statistical regression modeling setting. Principal component analysis, traditional correlation analysis, and multiple linear regression analysis are the three fundamental analytical methods combined in PLSR.

The SVM algorithm is a supervised machine learning model that employs non-linear mapping to map data in a high-dimensional data feature space. In a high-dimensional feature space, this enables the formulation of appropriate linear regression characteristics between the independent and dependent variables [45], enabling fitting in the higher-dimensional space and the subsequent return to the initial space. The core of the SVM regression model lies in the selection of the kernel function. In this study, several kernel functions' model prediction capabilities were analyzed before settling on the radial basis function (RBF). The SVM is widely used in the prediction of soil component composition using spectral data due to its high stability and generalization in solving non-linear issues [13,46].

The coefficient of determination ($R^2$), the mean absolute error (MAE), and the root mean square error (RMSE) were used to evaluate the accuracy of the prediction model. Lower MAE and RMSE values and higher $R^2$ values correspond to more precise model estimation [47].

Studies have shown that the d-factor can be used to evaluate the uncertainty of an estimation model [19,48]. The degree of uncertainty of an estimation model is proportional to the calculated value of the d-factor; that is, the degree of uncertainty of the estimated model will increase with the increase in the calculated value of the d-factor, and the d-factor calculation formula is as follows:

$$\overline{d_r} = \frac{1}{m}\sum_{i=1}^{m}(P_{Ui} - P_{Li}) \tag{10}$$

$$\text{d} - \text{factor} = \frac{\overline{d_r}}{\sigma_P} \qquad (11)$$

where $\underline{m}$ is the sample number; $P_{Ui}$ and $P_{Li}$ are upper and lower confidence limits, respectively; $\overline{d_r}$ is the average distance between the upper and lower confidence limits; and $\sigma_P$ is the standard deviation of the measured value of TN.

## 3. Results

### 3.1. Statistical Description of the TN Content of the Sampling Points

The 595 soil samples in this study were used, and for the train_test_split module in the Python programming language the random_state parameter was set to 15, and the soil samples were randomly divided into training and test sets at a ratio of 4:1. The numbers of training set and test set samples were 476 and 119, respectively, and the sample distribution of the training set and test set is shown in Figure 1c. The statistics of the samples that were taken are shown in Table 2.

**Table 2.** Results of the statistical description of the TN content.

| Set | N | Max (g/kg) | Min (g/kg) | Mean (g/kg) | SD (g/kg) | CV |
|---|---|---|---|---|---|---|
| Whole set | 595 | 1.80 | 0.37 | 1.01 | 0.22 | 0.22 |
| Training set | 476 | 1.80 | 0.37 | 1.01 | 0.22 | 0.22 |
| Test set | 119 | 1.71 | 0.43 | 1.02 | 0.24 | 0.23 |

### 3.2. TN Content Spectral Features Analysis and Spectral Transformation Processing

3.2.1. Spectral Features Analysis of TN Content

The ZY1-02D/AHSI hyperspectral reflectance spectral patterns of soil samples in the research area with various TN contents are shown in Figure 4. The distribution of the spectral reflectance curves for the various content levels followed a similar pattern, as shown by the original reflectance curves in Figure 4a. The spectral reflectance had two absorption peaks in the wavelength ranges 1100–1400 nm and 1750–2000 nm. According to pertinent research, the primary cause of the absorption peaks at 1100–1400 nm and 1750–2000 nm in soil water is hydroxide ions [49]. Generally, the soil reflectance values decrease as the TN content increases.

3.2.2. Spectral Transformation Processing

As can be seen in Figure 4b, the overall trend of the inverse reflectance spectral curve was the opposite of the original reflectance spectral curve, with a distinct reflection peak in the 1750–2000 nm wavelength range. Although the reflectance trough between 1750 and 2000 nm was also evident in the case of the natural logarithm of the reflectance (Figure 4c), the reflectivity curve was comparable with that of the original reflectance. Compared with the original reflectance, the inverse reflectance and natural logarithm of the reflectance showed a flattening out of the reflectance values in the wavelength range, except for the reflectance and absorption peaks that were highlighted at specific wavelengths, which helped to reduce noise-enhancing characteristic spectral variables. Figure 4d shows the first-order derivative reflectance spectral profile, which had more pronounced absorption peaks at 1100 nm, 1850 nm, and 1950 nm and distinct reflection peaks at 1150 nm, 1380 nm, and 2000 nm compared with the three other kinds of reflectance values.

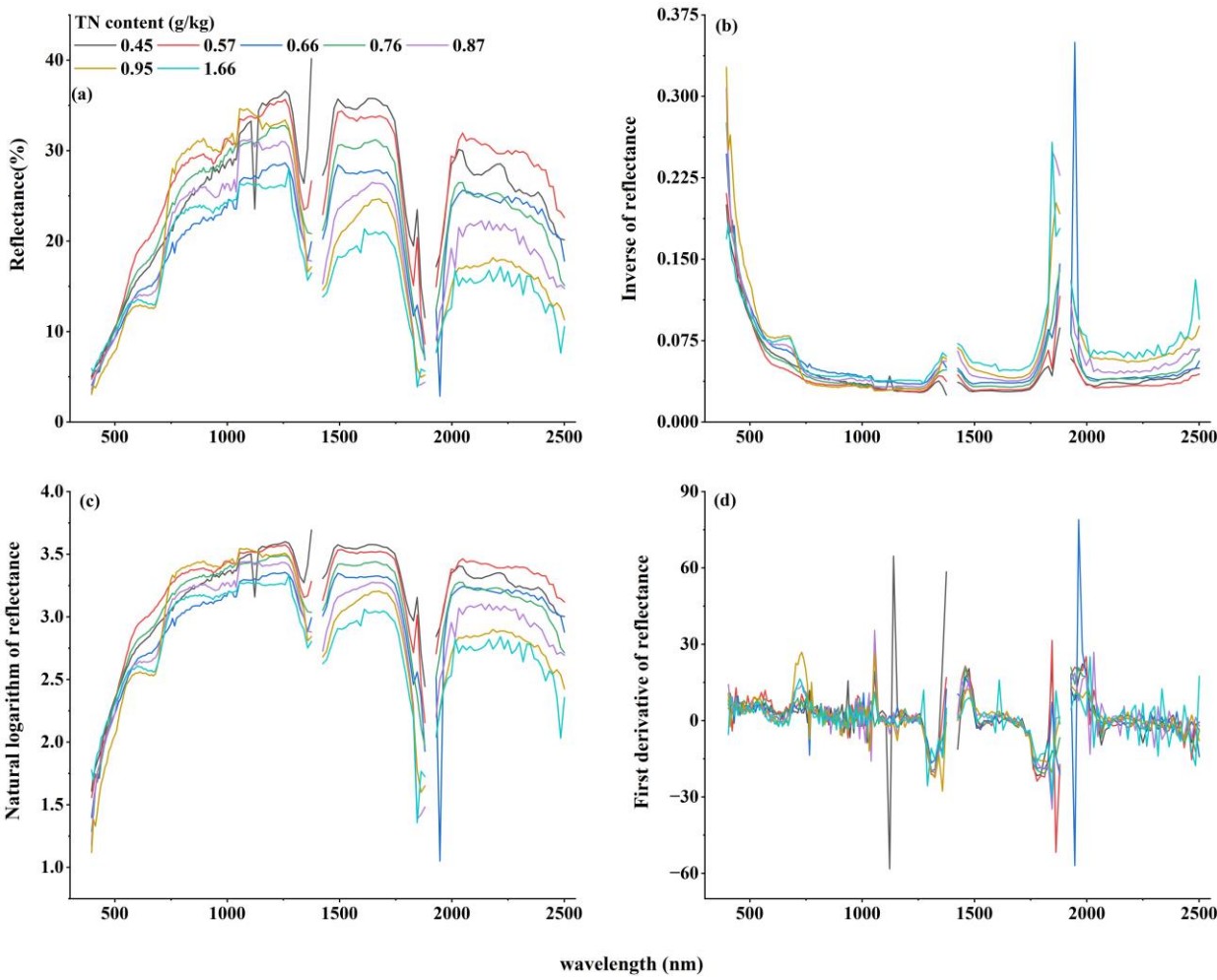

**Figure 4.** Spectral reflectance profiles for different TN contents: (**a**) original reflectance, (**b**) inverse reflectance, (**c**) natural logarithm of the reflectance, and (**d**) first-order derivative reflectance.

### 3.3. Selection of TN Spectral Characteristics

The LASSO algorithm was used to select the TN spectral bands based on the original and transformed reflectance readings. It can be seen from the results of the LASSO feature band selection in Figure 5 and Table 3 that the bands were much more distinctive for the original reflectance (OR) and first-order derivative reflectance (FDR) than for the inverse reflectance (IR) and natural logarithm of the reflectance (NLR). The main reason for this phenomenon was that the IR and NLR had prominent reflection and absorption peaks at specific wavelengths compared with the OR and FDR, and they had reflectance values that differed significantly from those in the other wavelength ranges. Therefore, the bands at the reflection and absorption peaks of IR and NLR were given larger coefficients when performing feature selection and were retained as feature variables, while the coefficients in other wavelength ranges were compressed to zero and excluded. The OR and FDR bands were evenly distributed between 400 and 2500 nm, while the IR and NLR bands were more concentrated, with most of the bands in the 400–700 nm and 1500–2000 nm wavelength ranges. The number of bands chosen for the LASSO feature is shown in Table 3 as n.

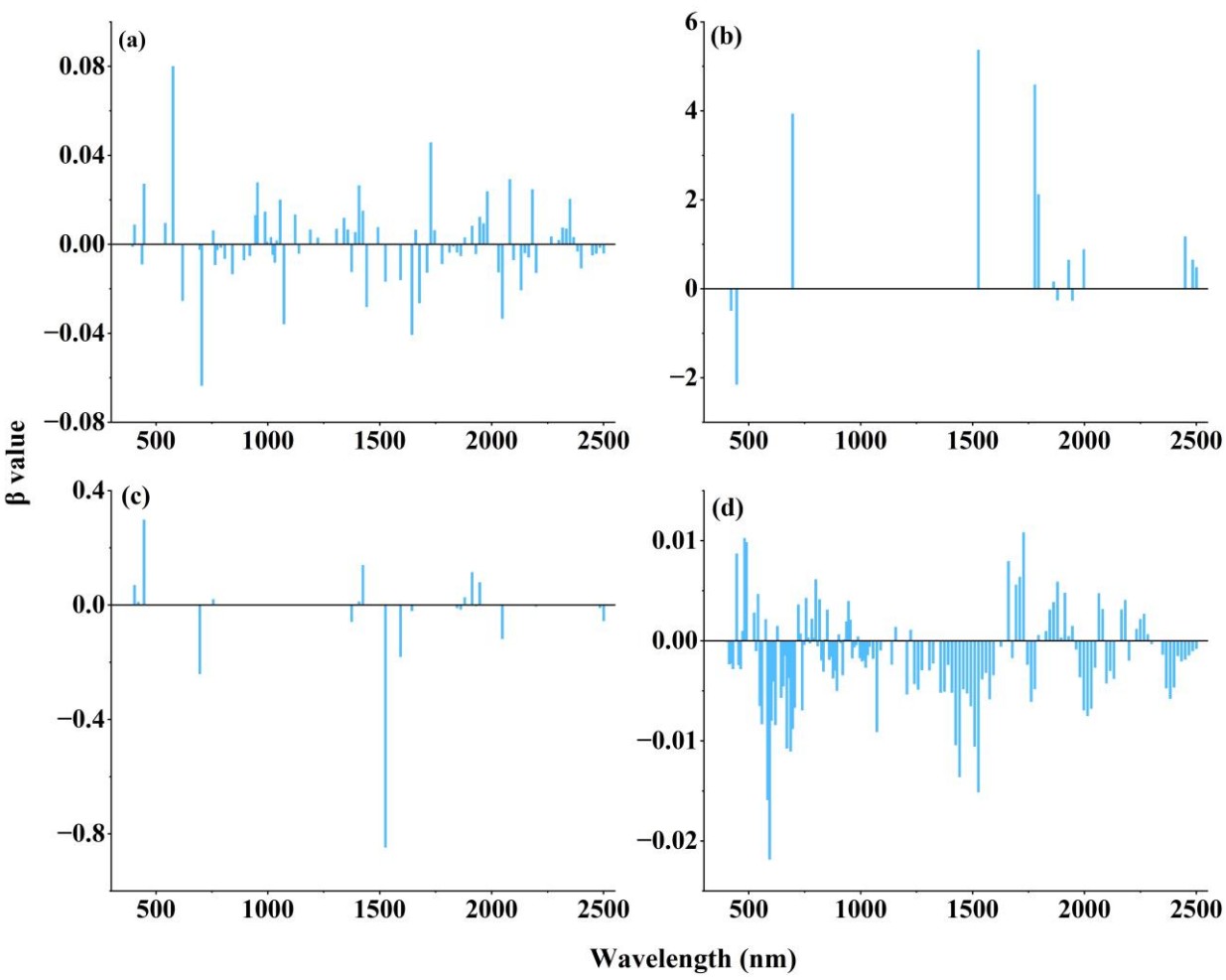

**Figure 5.** LASSO algorithm TN feature band selection results: (**a**) OR–LASSO, (**b**) IR–LASSO, (**c**) NLR–LASSO, and (**d**) FDR–LASSO.

**Table 3.** LASSO algorithm TN feature band selection results.

| Reflectance Representation | n | Wavelengths (nm) |
|---|---|---|
| OR | 77 | 396–405, 439–227, 542, 577, 619, 696–705, 756–774, 791, 808, 842, 894, 920, 945–954, 988–997, 1014–1073, 1123–1139, 1190, 1224, 1308, 1341–1442, 1493, 1526, 1594, 1644–1678, 1711–1745, 1779, 1812–1880, 1930–1981, 2014–2048, 2081–2098, 2132–2199, 2267, 2300–2401, 2450–2501 |
| IR | 17 | 395–404, 422, 447, 697, 1526, 1778–1795, 1845–1880, 1929–1947, 1998, 2451, 2484–2501 |
| NLR | 19 | 404, 422, 447, 697, 757, 1375, 1425, 1526, 1594, 1644, 1845–1880, 1930–1947, 2048, 2199, 2484–2501 |
| FDR | 141 | 404–430, 447–490, 524–559, 576–628, 645–705, 722–1106, 1139–1173, 1207–1274, 1307–1324, 1357–1594, 1627, 1660–1795, 1828–2132, 2165–2199, 2233–2317, 2350–2501 |
| Total | 254 | |

### 3.4. TN Content Model Estimation Results

3.4.1. Results of All Bands Based on the Individual Spectral Reflectance Transformation

MLR, PLSR, and SVM estimation models were formulated using the TN content as the dependent variable, and all bands of the four spectral transformations were used as independent variables. Table 4 presents the estimation results (OR, IR, NLR, and FDR in the table indicate the original, inverse, natural logarithmic, and first-order derivative

reflectances for all bands, respectively). MLR's training set accuracy was higher than that of PLSR and the SVM; however, its performance on the test set model was sub-par. The SVM model's prediction accuracy was better for both the training and test sets than that of PLSR, except in the case of first-order derivative reflectance. The test set performance of the SVM in terms of $R^2$ ranged from 0.45 to 0.59, the MAE ranged from 0.11 to 0.13 g/kg, and the RMSE ranged from 0.15 to 0.17 g/kg. SVM was the best model for estimating the TN content based on all bands after combining the data from the training and test set models.

**Table 4.** Estimation results of the TN content for all bands based on individual spectral reflectance transformations.

| Model | Reflectance | Training Set | | | Test Set | | |
|---|---|---|---|---|---|---|---|
| | | $R^2$ | MAE (g/kg) | RMSE (g/kg) | $R^2$ | MAE (g/kg) | RMSE (g/kg) |
| MLR | OR | 0.68 | 0.10 | 0.13 | 0.29 | 0.15 | 0.19 |
| | IR | 0.69 | 0.10 | 0.12 | 0.22 | 0.16 | 0.20 |
| | NLR | 0.69 | 0.10 | 0.12 | 0.28 | 0.15 | 0.19 |
| | FDR | 0.68 | 0.10 | 0.13 | 0.27 | 0.16 | 0.20 |
| PLSR | OR | 0.51 | 0.12 | 0.16 | 0.54 | 0.12 | 0.16 |
| | IR | 0.54 | 0.12 | 0.15 | 0.55 | 0.12 | 0.15 |
| | NLR | 0.53 | 0.12 | 0.15 | 0.54 | 0.12 | 0.16 |
| | FDR | 0.50 | 0.12 | 0.16 | 0.47 | 0.13 | 0.17 |
| SVM | OR | 0.60 | 0.10 | 0.14 | 0.59 | 0.11 | 0.15 |
| | IR | 0.61 | 0.10 | 0.14 | 0.58 | 0.11 | 0.15 |
| | NLR | 0.61 | 0.10 | 0.14 | 0.58 | 0.11 | 0.15 |
| | FDR | 0.58 | 0.10 | 0.14 | 0.45 | 0.13 | 0.17 |

The MLR model had the best results when the natural logarithm of the reflectance was used as the independent variable. The PLSR model with the natural logarithm of reflectance and inverse reflectance as inputs showed varying degrees of improvement in estimation accuracy compared with the original reflectance, with the natural logarithm of the reflectance having the best predictive power. OR–SVM was the best combination of models for the TN content estimation for all bands based on individual spectral reflectance conversions (Figure 6): $R^2$ was 0.59, the MAE was 0.11 g/kg, and the RMSE was 0.15 g/kg for the test set.

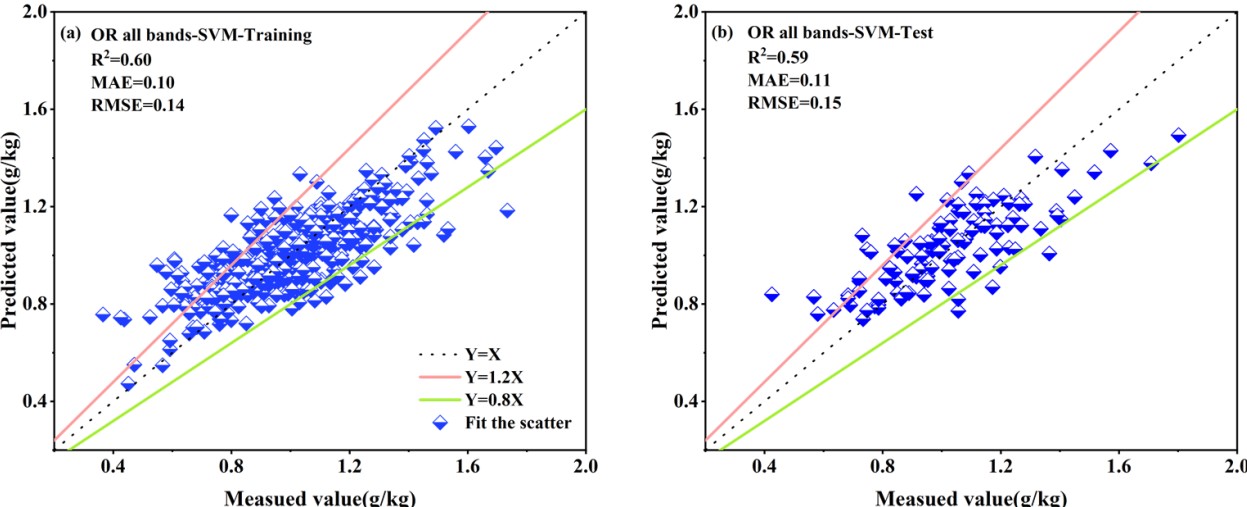

**Figure 6.** Scatter plots of measured and predicted TN contents based on individual spectral reflectance conversions of the best estimation model for all bands: (**a**) OR–SVM training set model; (**b**) OR–SVM test set model.

### 3.4.2. Results of the LASSO Feature Selection Based on the Individual Spectral Reflectance Transformations

Table 5 displays the results of the TN content estimate models' predictions, which utilized feature bands chosen with LASSO from the various spectral reflectance transformations. The outcomes of the training set models demonstrate that MLR and SVM had greater prediction accuracies than PLSR. The $R^2$ value of the MLR training set models ranged from 0.53 to 0.67, and higher $R^2$ values corresponded to lower MAE and RMSE values. The training set $R^2$ value for the SVM varied from 0.54 to 0.62, and the lowest values of MAE and RMSE were 0.10 and 0.14 g/kg, respectively. According to the outcomes of the test set models, the SVM model made the best predictions, followed by the PLSR and MLR models. The $R^2$ of the SVM for the test set ranged from 0.47 to 0.61, the MAE ranged from 0.11 to 0.13 g/kg, and the RMSE ranged from 0.14 to 0.17 g/kg. Combining the training and test set model results, the SVM was the optimal model for TN content estimation based on the LASSO feature selection.

**Table 5.** Estimation results of the TN content based on the LASSO feature selection.

| Model | Reflectance | Training Set | | | Test Set | | |
|---|---|---|---|---|---|---|---|
| | | $R^2$ | MAE (g/kg) | RMSE (g/kg) | $R^2$ | MAE (g/kg) | RMSE (g/kg) |
| LASSO–MLR | OR | 0.62 | 0.11 | 0.14 | 0.49 | 0.13 | 0.16 |
| | IR | 0.53 | 0.12 | 0.15 | 0.56 | 0.12 | 0.15 |
| | NLR | 0.54 | 0.12 | 0.15 | 0.55 | 0.12 | 0.15 |
| | FDR | 0.67 | 0.10 | 0.13 | 0.32 | 0.15 | 0.19 |
| LASSO–PLSR | OR | 0.55 | 0.12 | 0.15 | 0.54 | 0.12 | 0.16 |
| | IR | 0.52 | 0.12 | 0.15 | 0.57 | 0.12 | 0.15 |
| | NLR | 0.53 | 0.12 | 0.15 | 0.55 | 0.12 | 0.15 |
| | FDR | 0.51 | 0.12 | 0.16 | 0.48 | 0.13 | 0.17 |
| LASSO–SVM | OR | 0.58 | 0.11 | 0.14 | 0.61 | 0.11 | 0.14 |
| | IR | 0.54 | 0.12 | 0.15 | 0.56 | 0.12 | 0.15 |
| | NLR | 0.62 | 0.10 | 0.14 | 0.58 | 0.11 | 0.15 |
| | FDR | 0.57 | 0.11 | 0.15 | 0.47 | 0.13 | 0.17 |

A comparison of the different spectral reflectance transformations revealed that the inverse reflectance and natural logarithm of reflectance were the best inputs for the MLR and PLSR, respectively. With the SVM models, the natural logarithm of reflectance performed best in the training set, whereas OR performed best in the test set. OR–LASSO–SVM was, overall, the optimal model combination for LASSO feature selection to estimate TN content (Figure 7), with test set model metrics of $R^2 = 0.61$, MAE = 0.11 g/kg, and RMSE = 0.14 g/kg.

### 3.4.3. LASSO-Selected Spectral Band Combinations for the Four Spectral Reflectance Transformations

The four groups of LASSO characteristic bands of spectral reflectance transformation were combined as the input variables of the TN content prediction model. Table 6 displays the outcomes of the model estimations of TN content based on attributes collected from band combinations chosen by LASSO. The LBC–MLR training set model achieved better TN content estimation, with an $R^2$ of 0.80; nevertheless, when compared with the other models, the test set model did not perform as well (LBC: LASSO-selected spectral band combination). In terms of the test set accuracy, the LBC–PLSR and LBC–SVM models both outperformed the LBC–MLR, with the LBC–SVM model outperforming the LBC–PLSR model in terms of prediction outcomes. The LBC–SVM model, with test set model metrics of $R^2 = 0.57$, MAE = 0.12 g/kg, and RMSE = 0.15 g/kg, was the best model for TN content estimation among the LASSO-selected feature band combinations. The scatter plots of the LBC–SVM model's TN content measurements fitted to the predicted values are shown in Figure 8.

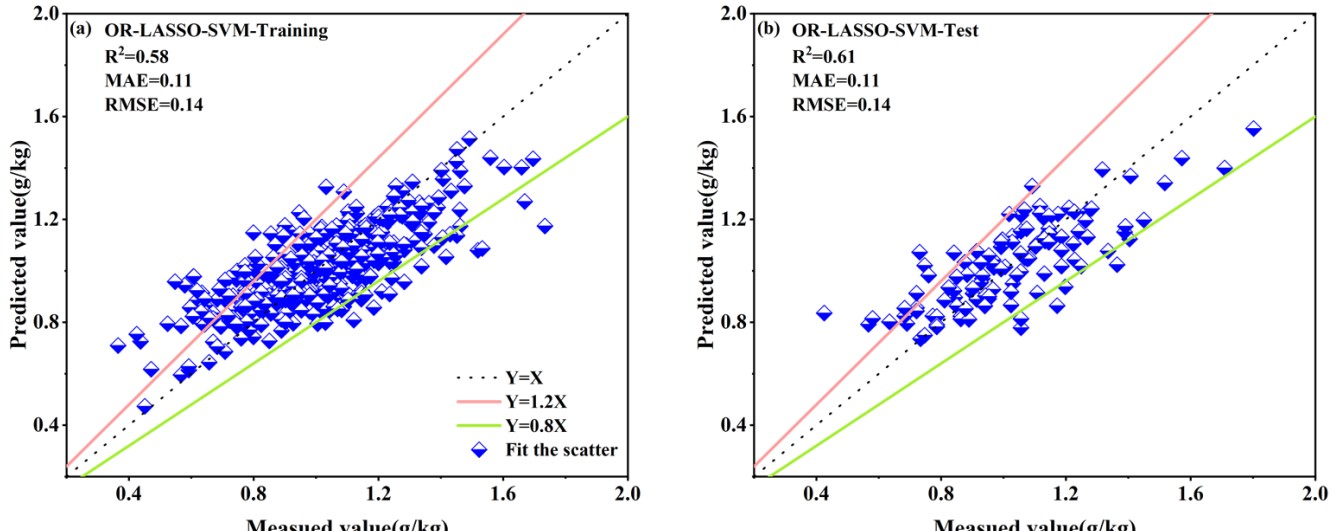

**Figure 7.** Scatter plots of the measured and predicted values of the best estimation model for TN content based on the LASSO feature selection: (**a**) OR–LASSO–SVM training set model; (**b**) OR–LASSO–SVM test set model.

**Table 6.** Estimation results of the TN content model based on the LBC.

| Model | Training Set | | | Test Set | | |
|---|---|---|---|---|---|---|
| | $R^2$ | MAE (g/kg) | RMSE (g/kg) | $R^2$ | MAE (g/kg) | RMSE (g/kg) |
| LBC–MLR | 0.80 | 0.08 | 0.10 | 0.23 | 0.19 | 0.24 |
| LBC–PLSR | 0.52 | 0.12 | 0.15 | 0.54 | 0.12 | 0.16 |
| LBC–SVM | 0.65 | 0.10 | 0.13 | 0.57 | 0.12 | 0.15 |

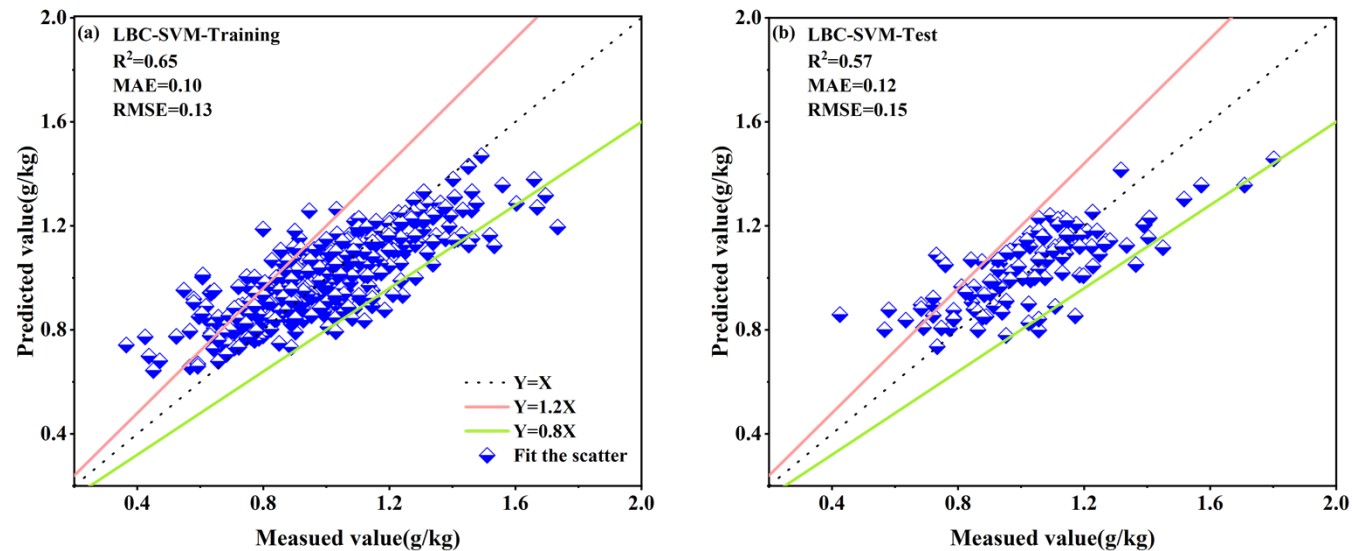

**Figure 8.** Scatter plots of the measured and predicted values of the best estimation model for TN content based on the LBC: (**a**) LBC–SVM training set model; (**b**) LBC–SVM test set model.

### 3.4.4. DE Secondary Feature Selection from LASSO-Selected Spectral Band Combinations Based on Four Spectral Reflectance Transformations

In this research, secondary feature selection from the LBC was carried out using the DE algorithm in combination with a prediction model. For the DE algorithm, 100 individuals were used and 1500 iterations were performed. In Table 7, the outcomes of the LBC–DE quadratic feature selection are displayed. LBC–DE–MLR was used to pick 98 variables,

LBC–DE–PLSR to select 61 variables, and LBC–DE–SVM to select 50 variables. N in Table 7 is the number of variables extracted.

**Table 7.** DE feature selection results for the LBC.

| Reflectance Representation | MLR Wavelengths (nm) | N | PLSR Wavelengths (nm) | N | SVM Wavelengths (nm) | N |
|---|---|---|---|---|---|---|
| OR | 404, 542, 576, 619, 765–774, 791, 954, 1023, 1056, 1139, 1190, 1308, 1442, 1526, 1644, 1745, 1812–1845, 1880, 1930-1947, 1981, 2031–2048, 2082-2098, 2183–2199, 2267, 2301, 2451 | 33 | 705, 757, 1880, 2132 | 4 | 447, 954, 1139, 1308, 1341, 1375, 1425, 1812, 2317, 2451 | 11 |
| IR | 396, 697, 1526, 1795, 1880, 1998, 2501 | 7 | 1526, 1880, 1930–1947, 2451, 2484–2501 | 7 | 404, 422, 1795, 1862, 2484, 2501 | 6 |
| NLR | 697, 757, 1425, 1526, 1880, 1930, 2199, 2484–2501 | 9 | 422, 1526, 1593, 1880, 1947 | 5 | 404, 4222, 1644, 1880, 1947 | 5 |
| FDR | 413, 490, 551–559, 594–628, 654–662, 679–688, 722–731, 757, 834, 851, 868–877, 894, 1073–1089, 1156–1173, 1257, 1375, 1442–1476, 1526, 1678-1728, 1762, 1795, 1947–1981, 2048, 2081, 2199, 2267, 2367–2417, 2451 | 49 | 413, 447–456, 551–559, 576-602, 619, 757, 808–842, 877, 894, 928, 946, 980, 1073, 1139, 1173, 1277, 1241, 1375, 1442, 1510–1526, 1678, 1728, 1829–1846, 1880, 1930–1947, 1981, 2048–2098, 2132, 2183, 2367 | 45 | 482, 525, 594–602, 619, 622, 679–688, 705, 748, 834, 877–885, 1073, 1089, 1224, 1526, 1745, 1778–1795, 1880, 1930, 2098, 2183, 2233, 2301, 2501 | 28 |
| Total | | 98 | | 61 | | 50 |

The LBC–DE–SVM model had an excellent ability to predict the TN content; its training set metrics were $R^2$ = 0.85, MAE = 0.08 g/kg, and RMSE = 0.09 g/kg, while the corresponding values for the test set were $R^2$ = 0.72, MAE = 0.08 g/kg, and RMSE = 0.12 g/kg (Table 8). The LBC–DE–MLR and LBC–DE–PLSR models were comparable in predictive ability but inferior to the LBC–DE–SVM model. The LBC–DE–SVM was the best model for TN content estimation based on the LBC–DE quadratic feature selection (Figure 9). Moreover, compared with Figure 6, Figure 7 Figure 8, the fitted scatter plots of both the training set and test set models in Figure 9 were closer to the 1:1 line. The results show that the LBC-DE-SVM model had a better TN content estimation ability than the other models.

**Table 8.** TN content model estimation results based on LBC–DE.

| Model | Training Set | | | Test Set | | |
|---|---|---|---|---|---|---|
| | $R^2$ | MAE (g/kg) | RMSE (g/kg) | $R^2$ | MAE (g/kg) | RMSE (g/kg) |
| LBC–DE–MLR | 0.64 | 0.10 | 0.13 | 0.65 | 0.10 | 0.14 |
| LBC–DE–PLSR | 0.57 | 0.11 | 0.15 | 0.60 | 0.11 | 0.14 |
| LBC–DE–SVM | 0.85 | 0.08 | 0.09 | 0.72 | 0.08 | 0.12 |

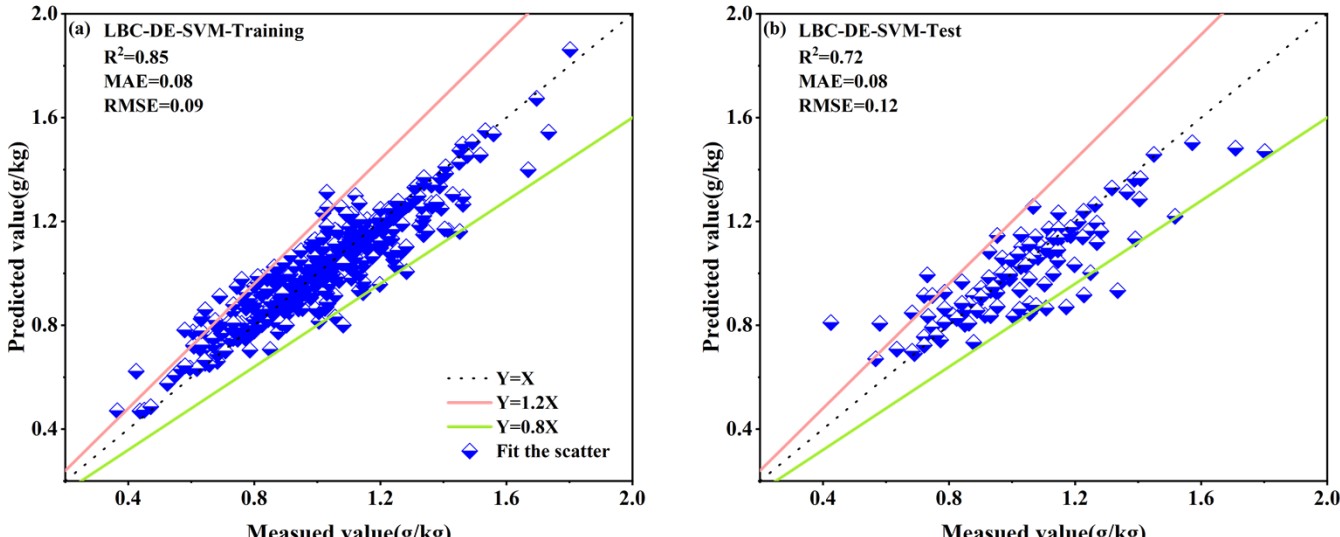

**Figure 9.** Scatter plot of the measured and predicted values of the best estimation model for TN content based on LBC–DE: (**a**) LBC–DE–SVM training set model; (**b**) LBC–DE–SVM test set model.

## 4. Discussion

### 4.1. Role of Spectral Reflectance Transformation Processing

In modeling the soil elemental content using hyperspectral data, an essential step is the mathematical transformation of the data. Spectral reflectance transformation can help to increase the precision of the prediction by lowering the interference from noise in the model [50,51]. In this study, the inverse reflectance and the natural logarithm of the reflectance enhanced the reflection peaks between 1750 and 2500 nm, and the first-order derivative reflectance transformation highlighted the absorption peaks at 1100 nm, 1850 nm, and 1950 nm and the reflection peaks at 1150 nm, 1380 nm, and 2000 nm. However, research demonstrated that not all spectral transformation techniques can produce outcomes superior to those obtained using the original data [16]. Regarding this, it should be mentioned that the best spectrum transformation varied depending on the model and even had an impact on how well a model performed on both the training and test sets. Using the prediction outcomes of the SVM model for LASSO feature selection based on the individual spectral reflectance change as an illustration, the training set model had the best accuracy when the natural logarithm of the reflectance spectral data was used as the input, but the test set model with the best estimation capability was that utilizing original reflectance spectral data. As a result, in this research, DE secondary feature extraction and modeling of the LASSO feature selection band combination (LBC) were carried out for the four spectral reflectance transforms in order to completely utilize the spectral reflectance transformation and to improve the prediction accuracy. Compared with the individual spectral reflectance, the accuracy of all models based on the LBC–DE quadratic feature selection was improved to different degrees, according to the model results (Table 8). This shows that the LASSO feature band combination data contained more essential information than the individual LASSO feature bands, and extracting this information from the combined spectra for model estimation could improve the models' predictive power.

### 4.2. The Role of Spectral Feature Extraction

Research demonstrated that using feature bands for modeling soil elemental content can lower the dimensionality of the data, make the prediction process simpler, and enhance the prediction outcomes [52,53]. Moreover, in the quantitative inversion of lake water depth and lake mineral content based on multispectral satellite remote sensing data, some scholars used the reflectance transformation combination data and the data after feature extraction as input variables for model estimation and achieved good estimation results [54–56].

Therefore, in this study, the LASSO algorithm was first used to screen the TN-predicting feature variables from the individual spectral reflectance representations to formulate a prediction model. Then, the DE algorithm was used to perform secondary feature selection and TN content estimation for the LASSO feature spectral band combination (LBC). The results from this study's test set model demonstrate that (see Figure 10), except for individual models, the LASSO-feature-selection-based estimation models had higher accuracies than the all-bands-based models (best $R^2$ value using LASSO feature selection: 0.61; best $R^2$ value of models using all bands: 0.59). This was mainly because the LASSO feature selection method could extract the characteristic spectral bands of TN according to the subtle relationship between TN and spectral reflectance and then achieve the purpose of eliminating redundant spectral variables and estimating the accuracy using the model. The model prediction results based on LBC–DE were noticeably superior to those based on the LBC, as shown in Figure 9 (best $R^2$ value for LBC–DE: 0.72; best $R^2$ value for LBC: 0.57). This implies that the LBC contained a significant amount of redundant information. The DE technique eliminated non-essential variables and retained valuable information, which enhanced the model's estimation capacity.

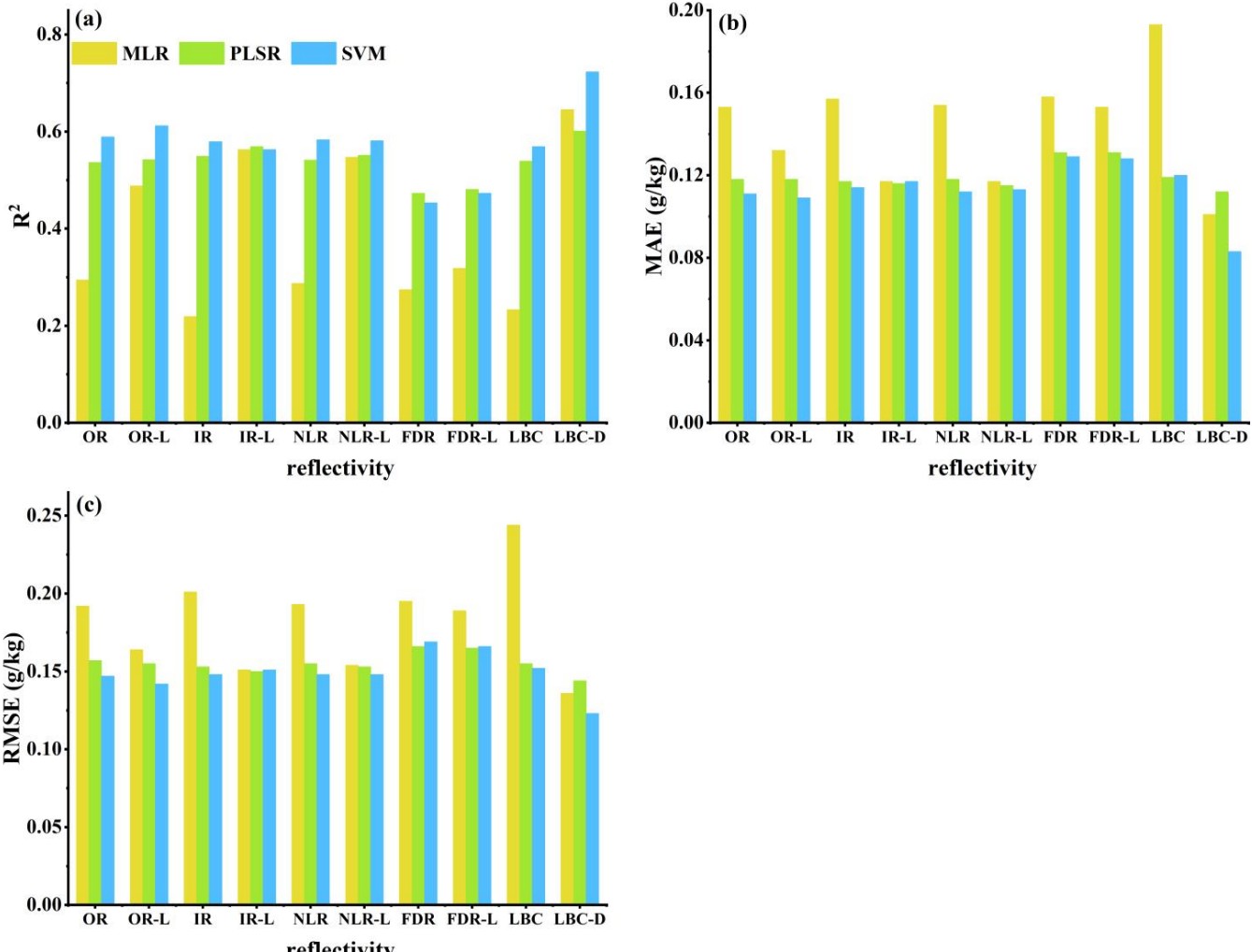

**Figure 10.** Comparison of the accuracy of the test set models: (**a**) $R^2$, (**b**) MAE, and (**c**) RMSE.

### 4.3. Estimated Model Comparison and Best Model TN Content Mapping

According to the comparison of the estimation accuracy and uncertainty evaluation indicators (Table 9) of the MLR, PLSR, and SVM models, the estimation accuracy of the SVM model was better than that of the MLR and PLSR models, and for the uncertainty calculation

of the SVM model training set and test set, the index d-factor value was comparable with the MLR and PLSR models. Overall, the SVM model had a better estimation ability for TN content. An SVM is a machine learning model used for solving non-linear problems [13,44] and can accurately estimate TN content and capture the link between TN content and complex spectral data. According to this study's findings, the accuracy of the SVM model was increased when the DE technique was applied to extract the feature variables from the LBC. The LBC–DE–SVM model was the best TN content estimation model in this study. As a result, the LBC–DE–SVM model was utilized to calculate the TN content of the research region's farming regions (Figure 11). There was a total of 692,389 image elements in the arable area of the study area, i.e., a total of 62,311.894 hectares of arable land, and the estimated map raster occupied 45.813 MB of memory per hectare. The TN concentration ranged from 0.57 to 1.52 g/kg in the study region, with a trend of being higher in the west and lower in the east.

**Table 9.** Calculation value of the estimated model d-factor.

| Model | Training Set | Test Set | Model | Training Set | Test Set |
|---|---|---|---|---|---|
| OR–MLR | 0.014 | 0.032 | FDR–LASSO–MLR | 0.014 | 0.033 |
| IR–MLR | 0.014 | 0.034 | OR–LASSO–PLSR | 0.013 | 0.029 |
| NLR–MLR | 0.014 | 0.033 | IR–LASSO–PLSR | 0.012 | 0.028 |
| FDR–MLR | 0.014 | 0.033 | NLR–LASSO–PLSR | 0.012 | 0.028 |
| OR–PLSR | 0.012 | 0.027 | FDR–LASSO–PLSR | 0.012 | 0.025 |
| IR–PLSR | 0.012 | 0.028 | OR–LASSO–SVM | 0.012 | 0.027 |
| NLR–PLSR | 0.012 | 0.028 | IR–LASSO–SVM | 0.012 | 0.027 |
| FDR–PLSR | 0.012 | 0.025 | NLR–LASSO–SVM | 0.013 | 0.028 |
| OR–SVM | 0.012 | 0.028 | FDR–LASSO–SVM | 0.010 | 0.021 |
| IR–SVM | 0.012 | 0.028 | LBC–MLR | 0.015 | 0.041 |
| NLR–SVM | 0.012 | 0.028 | LBC–PLSR | 0.012 | 0.028 |
| FDR–SVM | 0.010 | 0.021 | LBC–SVM | 0.012 | 0.025 |
| OR–LASSO–MLR | 0.013 | 0.031 | LBC–DE–MLR | 0.014 | 0.031 |
| IR–LASSO–MLR | 0.012 | 0.027 | LBC–DE–PLSR | 0.013 | 0.028 |
| NLR–LASSO–MLR | 0.013 | 0.028 | LBC–DE–SVM | 0.015 | 0.030 |

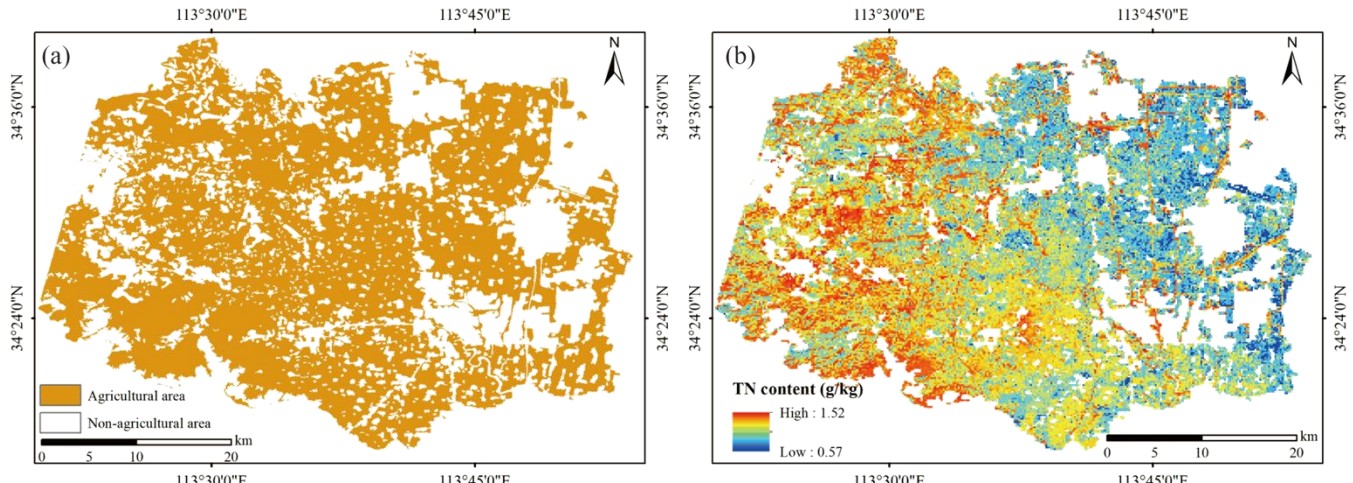

**Figure 11.** Map of the best model (LBC–DE–SVM) for estimating the TN content in the agricultural areas of the study area: (**a**) agricultural distribution in the study area and (**b**) spatial distribution of the TN content.

## 5. Conclusions

In this work, the ZY1-02D/AHSI hyperspectral satellite's remote sensing data and the measured soil sample data were used to estimate the TN concentration in the study region. In order to achieve an efficient and precise estimation of the TN content, the inverse

reflectance, natural logarithm of the reflectance, and first-order derivative reflectance were selected for the transformation of the original reflectance remote sensing data. The LASSO method was used to select the feature bands for the four sets of spectral reflectance transformations, and the combination of the DE algorithm and the prediction model was proposed to conduct secondary feature extraction and model estimation of the TN content for the LASSO feature band combination (LBC). The key conclusions that can be made include the following:

(1) The transformation of the spectral reflectance data can highlight some of the enhanced spectral information. However, the best spectral data pre-processing methods for different estimation models differ, where even the optimal spectral transformation methods for the training and test sets of the same model are different. Suitable spectral reflectance transformation methods can be selected for different prediction models in the TN content estimation studies in other regions to improve the estimation accuracy.

(2) Using the LASSO method for feature variable selection for full-band data not only reduced the spectral data redundancy and simplified the model but also improved the estimation accuracy of the model. Compared with individual spectral reflectance data, the LBC contained more valid spectral information and concentrated a large amount of noise information. This study used a combination of the DE algorithm and the prediction model to extract feature variables from the LBC, which can achieve the purpose of retaining valid information in the LBC and eliminating invalid information and can provide a reference for future research in making full use of the spectral reflectance transform and feature data for TN content estimation.

(3) Compared with ground-based hyperspectral data and airborne hyperspectral data, ZY1-02D/AHSI hyperspectral satellite image data have the advantages of wide image coverage, the automatic acquisition of hyperspectral remote sensing image data, and a short return cycle, and thus, it can enable the dynamic, rapid, and large area estimation of TN content.

Although the present study achieved satisfactory TN estimation in the central agricultural region of Henan Province, there were still some shortcomings, and further studies are needed to improve the existing results. On the one hand, there is a need for long-time-series monitoring of TN content based on the ZY1-02D hyperspectral data; in this study, only a single period of hyperspectral image data was used to establish a relationship with the soil total nitrogen content for the estimation study, and a study of the change in TN content over time and its correspondence with the change in spectral reflectance values of the images in different periods was missing. Therefore, to achieve long-time-series monitoring of TN content, there is still a need to further investigate the variation pattern of TN content in a long time series, as well as the correspondence between TN content and image spectral data in a long time series. On the other hand, this study was an estimation study based only on the relationship between spectral data and TN content, while TN levels in agricultural areas may be affected by the soil's physical and chemical properties, the temperature, the amount of moisture, the fertilizer application, and other factors. Therefore, various factors related to the TN content should be added to the estimation model in subsequent studies to improve the reliability of the model estimation.

**Author Contributions:** Writing—review and editing, R.Z. and J.C.; validation, W.Z.; methodology, D.Z.; data curation, W.D.; conceptualization, H.G.; project administration, S.Z. All authors have read and agreed to the published version of the manuscript.

**Funding:** This research was funded by Major Science and Technology Special Projects in Henan Province (221100210600), Major Science and Technology Special Projects in Henan Province (201400211000), Major Science and Technology Special Projects in Henan Province (201400210100), and the Science and Technology Tackling Plan of Henan Province (222102320220).

**Data Availability Statement:** Not applicable.

**Conflicts of Interest:** The authors declare no conflict of interest.

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
