# Peer review of "Estimation of the Total Soil Nitrogen Based on a Differential Evolution Algorithm from ZY1-02D Hyperspectral Satellite Imagery"

_agronomy, doi:10.3390/agronomy13071842_

Round 1
Reviewer 1 Report
The paper is in general interesting and supported by robust experimentation.
Nevertheless I would recommend the authors the following improvements before publication.
MAIN COMMENT: In march in China, what is the growth stage of wheat?
In other words, did the satellite took an image of naked soil or of soil covereed by wheat? In the scond case the paper is not on soil but rather on vegetation (which is of course proportional to nitrogen content). This is issue is changing completely the perspective of the work and must be carefully addressed in the paper.
If the satellite is including (completely or partially) the wheat vegetation, than the authors should better consider different varieties (with different growth speed), and different management practices.
Also presence of drains might affect mositure and thus response.
Othere comments:
In the paragraph "2.2. Soil sample acquisition and STN content determination" the authors should better describe which other parameters they have been collecting: SOM, STN and what else? Moisture? ph? salinity? Please add in the paper.
In general it would be good to avoid salami slicing and having in one paper different properties, rather than one paper per soil charactersitic.
Compared to the previous paper (doi: 10.3390/agronomy12092111) the sampled area has slightly changed: why did you decide to add remove positions?
I believe it would be useful to have the hyperspectral signal averaged on a couple of dates or at least to see how the performance s decreasing as the weather conditions are not clear enough: would it be possible to add results from another date? Please add in the paper.
Please provide the "digitization footprint" of the approach (expressed as Mb/ha or Gb/ha of the used images). Please add in the paper.
Try to avoid repetitions from the previous paper.
Some days were needed in order to collect 595 soil samples: what is the repeatability of soil analysis. Or in other words what is the effect of the time span on soil analysis and what is soil analysis repeatability (or uncertainty)?. Or again in other words: did you repeated soil sampling in one position after one or two weaks or in other 5 different close positions?
Authors mention the main crop (not the unique) is wheat: so what about other crops during sampling and satellite imaging? How could you manage the presence of different crops?
When submitting the revisede version, please highlight with yellow color in the manuscript the new corrected or added sentences.
English is not bad
Author Response
Dear editor and reviewer,
Thank you for your comments on our manuscript entitled " Soil Total Nitrogen Content Estimation Based on the Differential Evolutionary Algorithm for Secondary Feature Extraction with ZY1-02D Hyperspectral Satellite Imagery " (Manuscript ID: agronomy-2458683). Those comments are very helpful for revising and improving our paper, as well as the important guiding significance to other research. We have studied the comments carefully and made corrections which we hope meet with approval. The main corrections are in the manuscript and the responds to the reviewers’ comments are as follows (the replies are highlighted in red ). Please refer to the attachment for specific reply.

Reviewer 2 Report
This work fundamentally tries to estimate the local contents of Nitrogen in soil (and not the Satellite Telecommunication Network represented by STN as clearly identified by google search). It is based on several representation models, such as the multiple linear regression the partial least squares regression and a support vector machine the characteristics of which are operational on all bands, feature-bands, feature-band combinations, and the Differential Evolution algorithm feature band combination secondary feature variables. A single hyperspectral image retrieved from the ZY1-02D with the AHSI sensor for spectral reflectance transformation processing, spectral band selection/satellite and model construction methods obtained on January 28, 2021with as surface resolution of 30x30m2 and a revisit period of 3 days.
The authors found that spectral reflectance transformation can help increase the precision of the prediction by lowering the interference of noise on the model. They also coclosed that the data with Least Absolute Shrinkage and Selection Operator featured more essential information than the individual LASSO feature bands, and are suitable for extracting the Nitrogen contents from the combined spectra and are closer to several in-situ measurements for the sites used in predictive tests.
The aforementioned text should be used as highlight by the authors in order to revisit and reformulate completely their abstract which is currently in adequate and not representative of the authors’ work. Also, there they should mention the important work carried out with local measurements the number of which is impressive at the central Henan Province of China. Also, at the abstract should be mentioned why this region of the world was utilized for the tests carried.
In rewriting this manuscript, I would like to emphasize to the authors that they should enhance the generalization of their work in other geographical areas. For example, they could look in the literature and find suitable soil measurements of Nitrogen and take a similar image for this site in order to verify that this approach will be suitable to predicting Nitrogen contents elsewhere. Hence (the lines 506 to 513), should be appropriately expanded to include if such a generalization will be possible.
Also, at the conclusions, it should be enhanced the information (lines 525 to 527) with regards to the added value of this work, explain how the rapid estimation will be achieved over large areas (line 529). How rapid and how reliable this will be? The shortcomings mentioned (in line 531) should be specifically mentioned. Finally, at the conclusions (the lines 533 to 540, which are not supported by evidence in the main body of the text should be completely revised with sufficient evidence.
I am also happy to report that an important feature of the authors work is the text at lines 431 to 441. These topics are also worth mentioning at the concluding section and as a summary at the abstract.
Similar to the pervious comment are the section (lines 468 to 471).
On the contrary, the readers’ guidance section (lines 324 to 333) and the figure Fig.1c should be substituted with Fig.4. Thus, saving all necessary space for eliminating the acronyms (see the recommendations for the English) and enhancing the section of conclusions (as described above).
Another important element during the revision process is the number of significant digits after the decimal point. This is fundamental problem because these digits should have a significance only by the “accuracy of the process”. For example, at the legend of figure 5 appears 3 decimal digits for the g/kg. Do the authors know such an instrument that can be so accurate and sensitive to measure up to 1 thousand of g/kg of soil? The same is also true for the legend of Fig.12c for the total Nitrogen content.
Similar to the above, three decimal digits are not essential for the R2 (at lines 34, 35 and in many other places until line 462). This is a statistical number and should be rounded accordingly to whatever digit it makes sense for the accuracy of this work. Also, the authors need to comment up to what value of R2 is considered to be sufficient for their work. This is normally 8% in other works (lines 360-361).
Concerning Figs.7, 8, 9 and 10, the authors include the solid black line of the linear approximation. In all of these figures it will be essential to incorporate the +-20% lines in order to show how many measurement points will be included within this uncertainty values.
At line 154 at Table.1, the revisit period of the satellite is sufficient short (i.e. 3 days). It is important that this element is highlighted and it should be considered that with this option the authors have a very good opportunity to verify the total Nitrogen content of this work with subsequent satellite images or even to demonstrate the seasonality of nitrogen content variation during an annual production period.
As a last minor change, I would like to invite the authors to explain at the caption of Fig.1, lines 124-125 what is the blue for the Fig.1a. Is it the Henan Province or something different?
Please note that the all the aforementioned changes should be incorporated in the text of a major revision before further consideration for publication in this important journal.
The title of this manuscript is long and non-descriptive. A viable alternative could be: “Estimation of the total soil nitrogen based on a differential evolution algorithm from a hyperspectral satellite image”.
The authors extensively use the acronym STN for soil total Nitrogen. Yet, in this work are not examined nitrogen in water or air. Hence, the wording for soil is sufficient to be introduced once only at the introduction. For the remaining two words the manuscript will become reasonable by using only “N” with the subscript “t or T” as is common in chemistry.
The authors make an excessive use of acronyms most of which are not necessary. They even start using their definition from the abstract, which a totally unethical process. In the revised manuscript all acronyms of up to three letters should be substituted with the full text. This will significantly improve the readability and the significance of this work.
The use of the excessive acronyms makes the authors talking in riddles. For example, at lines 349-350 at the caption of Tables 4, the authors need to specify the all bands and reflectance transformations for which should be the impossible text of: “MLR, PLSR and VSM comparison for OR, IR, NLR and FDR”. Obviously, this is not reasonable and realistic.
The authors use the term “scholars” (line 73 and 453) and “researchers” (line84) are these two different entities or the same type of academics?
The paragraph (lines 334 to 342) is not easy to read and understand. It should be properly revised and perhaps enhanced.
The same clarity revision is required for the caption of Fig.11.
Round 2
Reviewer 1 Report
The paper has been improved in agreement with my comments.
I believe the paper is now more readable and understandable, better supported by information and descripion: thus believe it is now accepetable for publication.
English is in general fine: only minor typos to be amended
Author Response
Dear Editors and Reviewers, Thank you for your comments on our manuscript entitled "Estimation of soil total nitrogen based on differential evolution algorithm from ZY1-02D hyperspectral satellite imagery" (manuscript ID: agronomy-2458683). These opinions are very helpful to the modification and improvement of our thesis, and have important guiding significance for other research. We have carefully studied these comments and made corrections which we hope will be approved. Major corrections are in the manuscript, and responses to the reviewers' comments follow (responses are highlighted in red). Please refer to the attachment for specific responses.

Reviewer 2 Report
The main role of the reviewer in this important journal is, among others, to act as a “neutral reader” in order to identify the points in the manuscript where specific actions should be taken for clarifying areas where rational doubts are raised from a “neutral” reading.
I think that the authors in their cover letter remember the POINT-12 that it was asked from them to address” ALL the aforementioned changes should be incorporated in the text before further consideration for publication”. This requirement does not mean that they will need to agree with all points raised by the reviewer. Yet, even if they do not agree they should insert their thoughts and their reply in the main body of the manuscript for making their text relevant to a “neutral observer”.
Alas, this was not done in this first revision. The following are 9 out of 18 items that were not addressed by the new version of the manuscript.
POINT-2; The surface resolution is an area, therefore the dimension should be given in m times m i.e. in m2 as I have already suggested by my original correction. The text 30x30m is simply scientifically wrong.
The same is also true for the unit given at Table 1. The spatial resolution is an area and not a linear unit.
Also, the real value of this work is “…the important work carried out with local measurements the number of which is impressive at the central Henan Province of China”. The authors have not mentioned this neither at the abstract nor at the key highlights and at the conclusions. Without these in-situ data this work is not relevant for any scientific journal.
POINT-3; The generalization text and why they authors did not carry it should be explicitly mentioned with the quoted reference should be “extensively” discussed in the main text. The few words inserted are not sufficient and certainly not sufficient because only account for “…other regions (in order to) improve the estimation accuracy”.
POINT-6; To a major extent I am happy with the changes in Fig.1. Yet the authors still seem to ignore what is the blue shape in Fig.1a. This should be duly described in the caption of the figure.
POINT-7; The authors are free to claim any international or national measuring method for the weight of N. However, they must report the “measuring specifications of this weight”, the instrument used and also include the refence of the apparatus used. Only then it will be justified to use decimal digits for each g/kg of soil.
POINT-8; Even at the reference quote only two decimal digits were used for the g/kg quantities. As for the statistical parameters R2, MAE and RMSE I am happy with the text that they indicate the accuracy of the approach yet to retain three decimal digits for these statistical parameters is beyond any comprehension. Certainly, the consistency with other parameters of the text is not a rational reason.
POINT-9; The +-20% lines have improved the evaluation of the reader. Yet, it was a very significant surprise to sea at Fig.9 that most points are contained within those two lines both during “training” and “testing”. Judging from the number of measurements this is an important outcome of this approach and should be dully discussed for the LBC-DE-SVM
POINT-10; This text was ignored in the main body of the manuscript.
POINT-11; It is not clear what do you mean with “…the blue color in Fig. 1(c) will no longer be Sections will be given special explanations.”. My recommendation is to write a suitable comment for the caption of Fig 1c saying that “the reflectance of buildings from the images are excluded”.
POINT-15; The authors have reduced to reasonable extent the acronyms. Yet, they have nor complied to °… start using acronym definitions from the abstract, which is a totally unethical process°.
For all the aforementioned reasons, I believe that the manuscript is incomplete and makes no sense without reading the contents of the cover letter(s). Therefore, I am suggesting to the editor, that this paper should be “rejected” from publication to the Agronomy journal.
Machine editing is not always giving good outcomes.
Author Response
Dear editor and reviewer,
Thank you for your comments on our manuscript entitled "Estimation of soil total nitrogen based on differential evolution algorithm from ZY1-02D hyperspectral satellite imagery" (manuscript ID: agronomy-2458683). These opinions are very helpful to the modification and improvement of our thesis, and have important guiding significance for other research. We have carefully studied these comments and made corrections which we hope will be approved. Major corrections are in the manuscript, and responses to the reviewers' comments follow (responses are highlighted in red). Please refer to the attachment for specific responses.
